# Cross-Protection of an Inactivated and a Live-Attenuated Lumpy Skin Disease Virus Vaccine against Sheeppox Virus Infections in Sheep

**DOI:** 10.3390/vaccines11040763

**Published:** 2023-03-29

**Authors:** Janika Wolff, Martin Beer, Bernd Hoffmann

**Affiliations:** Institute of Diagnostic Virology, Friedrich-Loeffler-Institut, Federal Research Institute for Animal Health, Südufer 10, D-17493 Greifswald-Insel Riems, Germany; janika.wolff@ceva.com (J.W.); martin.beer@fli.de (M.B.)

**Keywords:** capripox virus, sheeppox virus, SPPV, lumpy skin disease virus, LSDV, inactivated vaccine, Lumpyvax, live-attenuated vaccine, cross-protection

## Abstract

Sheeppox virus (SPPV) (genus Capripoxvirus, family Poxviridae) infections are a highly virulent and contagious disease of sheep with a high morbidity and mortality, especially in naïve populations and young animals. For the control of SPPV, homologous and heterologous live-attenuated vaccines are commercially available. In our study, we compared a commercially available live-attenuated lumpy skin disease virus (LSDV) vaccine strain (Lumpyvax) with our recently developed inactivated LSDV vaccine candidate regarding their protective efficacy against SPPV in sheep. Both vaccines were proven to be safe in sheep, and neither clinical signs nor viremia could be detected after vaccination and challenge infection. However, the local replication of the challenge virus in the nasal mucosa of previously vaccinated animals was observed. Because of the advantages of an inactivated vaccine and its heterologous protection efficacy against SPPV in sheep, our inactivated LSDV vaccine candidate is a promising additional tool for the prevention and control of SPPV outbreaks in the future.

## 1. Introduction

The genus *Capripoxvirus* consists of the three species, namely lumpy skin disease virus (LSDV), sheeppox virus (SPPV), and goatpox virus (GTPV) [1]. Capripox virus (CaPV) infections are responsible for serious losses in cattle, sheep, and goats, highly affecting both the global economy and the livelihoods of small-scale farmers [2,3,4,5]. Because of their significant economic impact, CaPV-induced diseases are reported to be the most serious poxvirus diseases in domestic animals [6,7] and are classified as notifiable diseases under the guidelines of the World Organization for Animal Health (WOAH/OIE) [8]. The transmission of LSDV occurs as mainly vector-mediated via blood-feeding insects [9,10,11,12] and possibly hard ticks [13,14,15]. Contrarily, the transmission of SPPV and GTPV is reported to be mainly through direct contact between infected and naïve animals, as well as aerosols [3,4,7,16,17]. Clinical course can range from subclinical though mild to acute [18]. Affected animals develop clinical signs such as fever, enlarged lymph nodes, respiratory symptoms (coughing and nasal discharge), and characteristic pox-like lesions of the skin [2,16,19,20,21,22,23,24,25,26]. In addition, animals may suffer from temporary or permanent sterility [27], excessive salivation [28], and secondary bacterial infections [27]. The morbidity and mortality of SPPV are highly variable, depending on host factors such as age, breed, and immune status and virus factors (e.g., strain, virulence, and pathogenicity) [17]. Whereas morbidity and mortality in the indigenous breeds of endemic areas are usually less than 10% [7,29], morbidities and mortalities up to 100% have been observed in naïve populations [3,17,29]. The control of CaPV infections mainly relies on the early detection of an outbreak, movement restrictions, slaughter of infected and in-control animals, and ring vaccinations [3,17,30].

For the control of CaPV infections, only live-attenuated vaccines are commercially available [3,31] that are based on field isolates that were attenuated via multiple passaging in cell cultures or on the chorioallantoic membrane of embryonated chicken eggs [32,33,34]. As all CaPVs share a common major antigen for neutralizing antibodies [3], not only vaccination with homologous vaccines, but also immunization with heterologous vaccines, is successfully performed in the field [35,36]. Recently, Hamdi et al. evaluated the cross-protective potential of a live-attenuated LSDV vaccine (LSDV-Neethling strain derivative) against SPPV challenge infection in sheep in comparison with a homologous vaccination with the live-attenuated Romanian SPPV vaccine. In their study, homologous vaccination resulted in the complete clinical protection of the sheep, whereas the heterologous LSDV vaccine induced only partial protection against SPPV challenge infection [36]. Additionally, GTPV vaccines have been reported to be usually highly protective against SPPV infections in sheep [7]. Although many live-attenuated vaccines, especially when used in a homologous way, confer protective immunity in vaccinated animals in the field [37], there are some disadvantages that have to be considered. Vaccine failure and the development of severe adverse effects similar to field infections have been reported for CaPV live-attenuated vaccines [22,37]. Furthermore, the usage of these vaccines is not authorized in countries free of the diseases due to both trade restrictions [3,38] and a missing strategy of differentiating infected from vaccinated animals (DIVA) using serological methods. Therefore, there are great inhibitions to carry out preventive vaccinations. In recent years, the focus has thus been on the development of other types of vaccines, e.g., inactivated vaccines against CaPV infections. Because of their non-replicating characteristics, these vaccines are generally safe [22,31,39,40]. Although it has been claimed by some capripox researchers that animals vaccinated with inactivated CaPV vaccines are less protected and induce only short-term protective immunity [7], promising results have also been obtained in the past. Already around 40 years ago, an inactivated SPPV strain was able to induce compete clinical protection in sheep after virulent challenge infection [41]. A study published in 2003 revealed protection for at least six months post-vaccination with an inactivated SPPV-based vaccine against SPPV challenge infection [42]. Some years later, in 2016, Boumart et al. published their results of an animal trial in which they compared a live-attenuated SPPV vaccine with an inactivated SPPV vaccine candidate. In this case, except for an increased body temperature for two days and hypersensitivity reaction at the inoculation site of the challenge virus, no clinical signs of sheeppox (SPP) could be observed in the animals that received the inactivated prototype vaccine [22]. Similar results could be observed for inactivated vaccines against LSDV in cattle. In this instance, different virus isolates in inactivated form were able to clinically protect cattle against a strong challenge infection [39,40]. However, sheep vaccinated with a heterologous inactivated GTPV vaccine were only partially protected against SPPV challenge infection [43], making cross-protection an interesting point in the evaluation of newly developed inactivated vaccines against CaPVs.

In our study, we examined the cross-protective immunity of two different LSDV vaccines in sheep: the commercially available live-attenuated Lumpyvax vaccine strain from the MSD Animal Health and our recently developed inactivated LSDV vaccine candidate based on the LSDV-Serbia field strain adjuvanted with a low-molecular copolymer adjuvant that is able to induce sterile immunity in cattle [40].

## 2. Materials and Methods

### 2.1. Animals

Twenty of three to six-month old sheep (Dorper) were housed in the facilities of the Friedrich-Loeffler-Institut, Insel Riems, Germany, under the biosafety level 4 (animals) condition. Sheep were allocated randomly into three groups: two vaccination groups each consisting of eight animals and one challenge control group of four animals. All of the animals were in good condition, clinically healthy, and negative for capripox virus DNA, as well as antibodies against capripox viruses at the start of the trial. All of the respective animal protocols were reviewed by a state ethics commission and were approved by the competent authority (State Office for Agriculture, Food Safety and Fisheries of Mecklenburg-Vorpommern, Rostock, Germany; Ref. No. LALLF M-V/TSD/7221.3-1-052/20 (approval date: 10 September 2020).

### 2.2. Vaccine Preparation

The inactivated vaccine was prepared, as previously described [40]. Briefly, the LSDV-Serbia field strain was propagated on the adherent baby hamster kidney cell line BHK-21 (kindly provided by Zoetis, Olot, Spain) in a serum-free medium. The harvesting of the virus was performed via freeze–thawing the infected cell culture. Virus titer before inactivation was 10^7^ cell culture infectious dose_50_ (CCID_50_)/mL, as shown by the titration of the virus suspension on Madin–Darby bovine kidney (MDBK) cells (FLI cell culture collection number CCLV-RIE0261). The virus was inactivated using binary ethylenimine (BEI) using a standard procedure. Here, 200 mM sodium thiosulphate and 0.1 M 2-Bromoethylamine hydrobromide (BEA) in 200 mM sodium hydroxide were prepared and the mixture was incubated for 1 h at 37 °C for a cyclisation reaction. Afterwards, 36 mL of LSDV-Serbia virus preparation was incubated with 4 mL of cyclic BEA (BEI) and incubated for 24 h at 28 °C. Afterwards, the inactivation reaction was stopped by adding 4 mL of 200 mM thiosulphate and careful mixing. The validation of the virus inactivation was performed by three passages of the inactivated virus on MDBK cells and pan capripox real-time qPCR analyses (see Section 2.4). Inactivated antigen was formulated with “Adjuvant A” [40] (10% Polygen, MVP Adjuvants^®^, Omaha, Nebraska, USA, batch No. P10061).

For the preparation of the live-attenuated vaccine, the Lumpyvax (MSD Animal Health, South Africa; LSDV SIS Neethling-type strain) life virus was propagated on an MDBK cell line with a cell culture medium + 10% FCS. The harvesting of the virus was performed by freezing and thawing the infected cell culture. Virus suspension was titrated on MDBK cells, and a titer of 10^8.2^ cell culture infectious dose_50_ (CCID_50_)/mL was determined. Before the vaccination of the animals, the virus preparation was diluted in TE puffer (pH 8.0) to a final titer of 10^5^ CCID_50_/mL, which was confirmed by back titration.

### 2.3. Experimental Design and Sample Collection

The animals of group A were vaccinated with the inactivated LSDV vaccine prototype. Vaccination was performed twice with primary immunization at day 0 of the animal study (equivalent to 0 day(s) post-vaccination, dpv) and secondary vaccination at 28 dpv. Each animal received 2 mL of the vaccine preparation subcutaneously. The sheep of group B were vaccinated subcutaneously with 2 mL of a commercially available LSDV vaccine strain (Lumpyvax, MSD Animal Health, Intervet, Spartan, South Africa) at day 28 of the animal study (28 dpv). The sheep of group C were not vaccinated and served as the challenge control group.

Challenge infection was performed 21 days after the last vaccination at day 49 of the animal study (49 dpv, 0 day(s) post-challenge, dpc). Thereby, all of the animals were intranasally inoculated with 2 mL of highly virulent SPPV-India/2013/Surankote strain [23] with a titer of 10^6^ CCID_50_/mL.

The vaccinated animals were monitored for adverse effects against the vaccines for 14 days after each vaccination. Body temperature was measured daily from −2 dpv until 28 dpc. In addition, clinical reaction after the experimental challenge infection was scored daily from 0 dpc to 28 dpc, using a modified clinical reaction score for LSDV infections in cattle [23,25] based on a score system of Carn and Kitching [20]. At defined time points (0 dpv as well as 0, 3, 5, 7, 10, 14, 21, and 28 dpc), EDTA blood was taken for the evaluation of viremia. In addition, nasal swab samples (CLASSIQSwabs™ Cat.No 155C; Copan, Brescia, Italy) were taken for the analysis of viral shedding on the same days. The serum samples were taken and used to examine the serological response towards both vaccination and challenge infection. During necropsy, a panel of different organ samples (cervical lymph node (ln), mediastinal ln, and lung tissue) was taken and analyzed regarding the viral genome load.

### 2.4. Molecular Diagnostics

For the homogenization of organ samples in a serum-free medium, the TissueLyzer II tissue homogenizer (Qiagen, Hilden, Germany) was used. Subsequently, the DNA of all samples taken during the study was extracted using the KingFisher Flex System (Thermo Scientific, Darmstadt, Germany) and the NucleoMag Vet kit (Macherey-Nage, Düren, Germany) according to the manufacturer’s instructions. During DNA extraction, an internal control-DNA (IC-2 DNA) was added to the samples to control successful DNA extraction and inhibition-free amplification [44,45]. The analysis of the viral genome load in the samples was performed using the PerfeCTa qPCR ToughMix (Quanta BioSciences, Gaithersburg, MD, USA) and the already described pan capripox real-time qPCR of Bowden et al. [6] with a modified probe [46].

### 2.5. Serological Examination

For the serological analyses, two different assays were used: a double antigen (DA) ELISA and the serum neutralization test (SNT).

The ID Screen Capripox Double Antigen ELISA (ID.vet, Montepellier, France) was performed following the manufacturer’s instructions.

For the SNT, the serum samples were heat-inactivated for 30 min at 56 °C. Subsequently, the samples were diluted 1:10 in a serum-free medium followed by the preparation of log_2_ dilution series in triplicates in a 96-well plate format. Afterwards, 50 µL of the LSDV-Neethling vaccine strain with a titer of 10^3.3^ CCID_50_/mL was added to each well. After incubation for 2 h at 37 °C and 5% CO_2_, approximately 30,000 MDBK cells/100 µL were added to each well, followed by an incubation step for 7 d at 37 °C and 5% CO_2_. Subsequently, the development of CPE was analyzed with the Nikon Eclipse TS-100 light microscope, and neutralizing titer was calculated using the Spearman and Kärber method [47,48].

## 3. Results

### 3.1. Adverse Effects after Vaccination

During the vaccination phase, body temperature was measured daily beginning from 3 dpv (vaccination groups) and 4 dpv (challenge control group). After vaccination with the inactivated LSDV vaccine candidate, all of the animals except one (S-311) showed an increased body temperature from around 40.0 °C to 40.6 °C for a single day. In the following days, the body temperature of individual animals increased slightly over 40.0 °C, but only for single days. This phenomenon could also be observed after boost immunization at 28 dpv; however, body temperatures increased to a lower extent (from around 40.0 °C to 40.4 °C) and at fewer days than after prime immunization. The highest temperature measured in this group during the vaccination phase was 40.7 °C (S-284, 29 dpv). The animals that were immunized with the live-attenuated vaccine were vaccinated at 28 dpv. However, an increased body temperature could also be seen before vaccination between 3 dpv and 28 dpv with body temperatures up to 40.5 °C in individual animals at single days. After vaccination at 28 dpv, the body temperature of all of the sheep remained in the normal range for all days of the vaccination phase. The highest body temperature of this group could be observed for S-299 and S-293 at 3 dpv. The animals of the unvaccinated control group showed the lowest increase in temperature during the vaccination phase compared with both other groups. Here, only very few animals had an increased body temperature on single days, and the highest measured body temperature was determined to be 40.2 °C (S-309, 31 dpv). Adverse effects other than an increased body temperature—for example, local reactions—could not be observed in any of the vaccinated animals.

### 3.2. Clinical Reaction after Experimental Challenge Infection

After challenge infection, the body temperature of all sheep vaccinated with the inactivated LSDV vaccine prototype remained in the normal range until the end of the study (Figure 1A). A similar pattern could be observed in the group vaccinated with the live-attenuated LSDV vaccine. Here, only two animals showed a very slight increase in body temperature at 1 dpc (S-287, 40.1 °C) and 4 dpc (S-300, 40.0 °C) (Figure 1B). In contrast, all four animals of the challenge control group showed an increased body temperature, beginning at 3 dpc (S-309, 40.2 °C), and with all animals having fever from 4 dpc onwards. In the following, the fever lasted for 13 (S-315) to 22 days (S-309), reaching values higher than 41 °C in individual animals (S-304 and S-315) at single days post-challenge infection (Figure 1C).

Clinical reaction was determined daily from 0 dpc to 28 dpc. Thus, the development of clinical signs typical for SPP was examined and the overall clinical reaction score was calculated. Whereas all of the vaccinated animals, independent of the used vaccine, did not show any clinical signs after challenge infection at all (Figure 2A,B), all four animals of the challenge control group developed SPP. Already at 5 dpc, a slight nasal discharge (S-304 and S-298), development of single skin lesions (S-309 and S-298), and respiratory signs such as coughing or labored breathing (S-298) could be observed, leading to clinical reactions scores between 0.5 and 1.5 in the respective animals (Figure 2C). In the following days, the clinical course of all animals became severe, with sheep showing a bad general condition, reduced feed intake, localized and generalized skin lesions, strong nasal discharge, and respiratory problems, leading to clinical scores between 6.5 (S-304) and 10.5 (S-298) at 12 dpc. Whereas three sheep recovered from SPPV infection until the end of the study, S-298 reached the human endpoint and had to be euthanized at 15 dpc (Figure 2C).

### 3.3. Virus Replication and Shedding

For the evaluation of viremia, EDTA blood was taken at defined time points after challenge infection and was tested regarding viral genome load. The EDTA blood samples of all of the animals vaccinated with the inactivated LSDV vaccine candidate scored negative for viral genome load at all of the tested time points (Figure 3A). A similar result could be seen for the animals vaccinated with the live-attenuated vaccine. Here, no viremia could be observed, with the single exception of S-300 at 5 dpc; however, the Cq value was comparably high (35.9), indicating only slight viremia (Figure 3B). In the challenge control group, viremia could be detected beginning from 3 dpc (S-315, Cq 32.7). At 5 dpc, the EDTA blood of all four animals of this group scored positive for viral genome load, with Cq values ranging from 36.5 (S-315) to 34.3 (S-304) and 33.9 (S-309) to 29.7 (S-298). S-298, which was removed from the trial at 15 dpc due to a severe clinical course, remained positive until the last sampling at 14 dpc. Whereas no viral genome could be detected in the EDTA blood of S-315 from 14 dpc onwards, the individual EDTA blood samples of S-304 and S-309 turned negative and scored positive again at a later time point. However, Cq values were comparably high at these late time points (around 37) (Figure 3C).

In addition to viremia, viral shedding via nasal fluid was examined after experimental challenge infection. In the inactivated vaccine group, viral shedding started at around 3 dpc (S-308, S-311, and S-288), with high Cq values at around 38. At 5 dpc, all animals of the inactivated LSDV vaccine group showed viral replication in the nasal tissue indicated by viral genome in the nasal swab samples. Whereas S-284 showed only slight nasal shedding for a short period of time (only at 5 dpc, Cq 35.6), some animals remained positive until 21 dpc (S-288, S-296) but with high Cq values. In contrast, Cq values between 22.5 (S-308, 10 dpc), 23.1 (S-308, 14 dpc), and 23.5 (S-311, 10 dpc) indicate viral replication to a higher extent in the respective animals (Figure 3D). Viral shedding could also be observed in all animals vaccinated with the live-attenuated LSDV vaccine. Here, nasal shedding started at 3 dpc with Cq values around 38 (S-299 and S-293) and 34 (S-300). In the following, all of the animals scored positive for viral genome in nasal swab samples at least once. Most animals started to show viral shedding at 3 dpc (S-299, S-293, and S-300) and 5 dpc (S-294, S-286, and S-310), with two exceptions. A nasal swab sample of S-305 scored positive at 10 dpc the earliest, and S-287 showed viral shedding as late as 21 dpc. The lowest Cq values could be detected for S-300 with Ca values of 21 (7 dpc) and 25 (5 dpc) (Figure 3E). Overall, nasal shedding was similar between both vaccinated groups (Figure 3D,E). In the challenge control group, stronger viral shedding compared to the vaccine groups could be seen. Two out of four animals started to shed the virus at 3 dpc (S-304, Cq 39.3, and S-298, Cq 36.4). At 5 dpc, all of the animals scored positive for viral genome in the nasal swab samples, with low Cq values around 22. The viral shedding of all of the unvaccinated control animals lasted until the end of the study and, in the case of S-298, until euthanasia. Compared with the vaccine groups, the viral genome load was higher in the nasal swab samples, which was indicated by lower Cq values. The peak of viral shedding was at 10 dpc, with Cq values around 18 (Figure 3F).

### 3.4. Viral Genome Load in Certain Organ Samples

During necropsy, the cervical lymph node, mediastinal lymph node, and lung tissue were sampled, and the viral genome load was determined. The animals vaccinated with either the inactivated LSDV vaccine candidate or the live-attenuated LSDV vaccine did not show any viral genome in any of the sample matrices. This finding could also be observed in the challenge control group for two out of four animals (S-304 and S-315). Solely S-309 scored positive in the cervical lymph node (Cq 34.6), and viral genome could be detected in the cervical lymph node and the lung tissue of the euthanized S-298 (Cq 34.4 and 31.4, respectively) (Table 1).

### 3.5. Serological Response

For the evaluation of the serological response, the DA Antigen ELISA from ID.vet as well as the SNT were used. Before vaccination, all of the animals of the vaccination groups were negative for antibodies against CaPVs. The animals of the challenge control group tested negative for CaPV-specific antibodies at the day of challenge infection. Seroconversion after vaccination with the inactivated LSDV vaccine candidate started after prime immunization. At 28 dpv, the day of boost immunization, S-311 was positive in the DA ELISA and four out of eight sheep (S-311, S-313, S-289, and S-296) already showed neutralizing antibodies in the SNT. On the day of challenge infection, the sera of all sheep of this group were positive in both serological assays. After challenge infection, the neutralizing titer increased markedly in three sheep (S-308, S-311, and S-313), whereas the level of neutralizing antibodies remained similar in the remaining sheep (Figure 4A,D). The serum samples of sheep vaccinated with the live-attenuated vaccine did not react in the DA ELISA at the day of challenge infection; however, two animals (S-293 and S-305) tested positive for neutralizing antibodies in SNT. During the challenge period, only three animals (S-287, S-300, and S-286) developed antibodies detected by the DA ELISA with seroconversion beginning at different time points (7 dpc, 14 dpc, and 28 dpc). The other five sheep scored negative in the DA ELISA during the entire study. At 7 dpc, neutralizing antibodies could be detected using the SNT for three out of eight animals (S-287, S-300, and S-293), and S-299 and S-286 turned positive beginning at 14 dpc. In general, most animals showed seroconversion at a level near the threshold of both assays, except S-300, for which a high level of neutralizing antibodies against CaPV could be observed. In contrast, the serum samples of S-310 did not show any neutralizing activity at all during the study (Figure 4B,E). The animals of the challenge control group did not show measurable seroconversion until 28 dpc in the ELISA and 14 dpc in the SNT. However, at 28 dpc (ELISA) and starting from 14 dpc (SNT), the sera of all four animals scored positive in the respective serological assays (Figure 4C,F). The neutralizing titers were higher than those observed for the live-attenuated LSDV vaccine group and comparable to the group of inactivated LSDV vaccine-candidate immunized sheep (Figure 4D–F).

## 4. Discussion

SPP is an acute and highly contagious transboundary viral disease affecting sheep as well as, in the case of certain virus strains, goats [49]. Next to quarantine, the slaughter of the infected herds, and movement restrictions, vaccination with live-attenuated vaccines is performed to control the spread of SPPV [3,17]. As all three species of CaPVs are antigenically related [28], and cross-protection against heterologous CaPV infections in the natural hosts have been reported frequently [7,32], both homologous and heterologous vaccines can be used for vaccination campaigns [35,36]. Because of their non-replicating characteristic, inactivated vaccines are known to be safe in the individual animal after administration [22,39,40]. Recombination events, as reported recently for a contaminated live-attenuated LSDV vaccine containing multiple viruses [50,51,52], are not possible in inactivated vaccines and any spread of vaccine virus cannot occur. Therefore, these vaccines provide a helpful tool for preventive vaccination against CaPVs and the control of CaPV outbreaks. In the last years, promising inactivated vaccine candidates against LSDV and SPPV have been developed and successfully tested in homologous challenge studies [22,39,40].

In our study, we examined the cross-protective efficacy of a commercially available live-attenuated LSDV vaccine strain (Lumpyvax) and of our inactivated LSDV prototype vaccine in sheep after experimental challenge with a highly virulent SPPV field strain.

After prime and boost vaccination with the inactivated LSDV vaccine, a slight to moderate increase in body temperature in almost all of the animals of this group could be observed on single days. However, animals that were not vaccinated at this time (e.g., animals vaccinated with the live-attenuated vaccine at 28 dpv and the unvaccinated controls) developed increased body temperature for single days to a similar extent. Therefore, a direct correlation between vaccination with the inactivated LSDV vaccine and an increase in body temperature can neither be confirmed nor rejected. Next to increased body temperature, local reaction at the site of inoculation could be observed in a few animals after vaccination with a heterologous live-attenuated LSDV vaccine in a previous study of Hamdi et al. [36]. In contrast, no local reaction after the administration of both LSDV vaccines could be observed in our study.

After experimental challenge infection with the virulent SPPV-India/2013/Surankote field strain, all sheep of the unvaccinated control group developed fever (Figure 1C) and clinical signs characteristic for SPP. In detail, respiratory signs such as labored breathing, coughing, and nasal discharge could be observed. Additionally, all four sheep developed typical pox-like lesions at the skin that were generalized in some cases. Whereas three of the four unvaccinated control sheep recovered from the infection, S-298 reached the humane endpoint and had to be euthanized at 15 dpc (Figure 2C). The observed clinical course is in line with previous findings after the intranasal inoculation of sheep with SPPV-India/2013/Surankote [23], and similar clinical signs are observed regularly in SPP-affected sheep in experimental infection studies with other SPPV strains as well as during field outbreaks [6,7,22,23,28,36], leading to the conclusion that the used challenge model is appropriate and robustly induces clinical SPP in inoculated sheep. In contrast with the unvaccinated control animals, all vaccinated animals, independent of the vaccine used, were completely protected against any clinical signs of SPPV infection. Body temperature remained normal over the entire challenge period (Figure 1A,B), and no clinical signs could be observed in all of the vaccinated sheep (Figure 2A,B). These results are consistent with data published by Boshra and colleagues [53]. After the vaccination of sheep and goats with an IL-10 gene deficient LSDV recombinant and subsequent challenge infection, no viraemia could be detected in the vaccinated animals either. Unfortunately, no nasal swabs were examined in the trial, so no statement can be made on the local replication of the challenge virus in the nasal region. Success in controlling sheep pox and goat pox has also been reported following the use of the Kenyan sheep pox and goat pox (KSGP) vaccine [4], which, according to recent sequencing analyses, is an LSDV [54]. Interestingly, these findings differ from the results obtained during the study by Hamdi et al. In that study, heterologous vaccination with a live-attenuated LSDV vaccine led only to partial clinical protection in sheep after challenge infection. In the respective study, an inconsistent clinical course was observed, ranging from almost complete protection, with sheep showing only a mild increase in the body temperature, to partial protection indicated by the development of skin papules and no protection at all [36]. A possible explanation for these differences in clinical protection between the study of Hamdi et al. and our study could be the vaccination dose. Whereas the sheep in the previous study were vaccinated with 10^3^ CCID_50_ [36], the sheep in our study received 10^5^ CCID_50_ of the live-attenuated vaccine used. These two log_10_ differences in the vaccine virus dose might have influenced the outcome of the challenge infection and possibly led to more efficient protection against SPPV. In addition to protection efficacy against clinical signs, we analyzed the viral genome load in the blood samples and nasal swab samples to obtain insight into viremia and viral shedding, respectively. Marked differences between both the vaccinated groups and the unvaccinated control group were also seen for viremia, but not for viral shedding. In contrast to the control sheep, which all were viremic beginning from 5 dpc to the end of the study, no viral genome could be detected in the blood of any of the sheep vaccinated with the inactivated LSDV vaccine (Figure 4A), and only one sheep of the live-attenuated vaccine group scored positive for viral genome in the blood in a single day (5 dpc) with a high Cq value (Figure 4B). Contrarily, viral shedding via nasal fluid could be observed in all sheep, regardless of whether they were vaccinated or not (Figure 4D–F). Moreover, the viral genome load in different organ samples taken during necropsy was analyzed. No viral genome could be detected in cervical and mediastinal lymph nodes, or in the lung tissue in any of the vaccinated animals and in two out of four control sheep, whereas the remaining two control animals scored positive in one (S-309, cervical lymph node) and two (S-298, cervical lymph node and lung) of the three organ samples (Table 1). We cannot exclude that the internal organs of the negative sheep were not infected at all, but only that the virus genome was not detectable in these animals at the time of necropsy. Taken together, these molecular data indicate that both of the vaccines were able to prevent a generalized infection and viral replication in the immunized sheep (Figure 4A,B), but the local replication of the challenge virus in the nasal mucosa after intranasal inoculation did occur even after vaccination (Figure 4D,E). However, it should also be clear that a very high dose of the virulent cell culture virus was used for the challenge infection. It is very likely that, in the course of a natural infection, comparable amounts of infectious virus would not reach the nasopharynx of a vaccinated contact sheep in a comparably short time. As no virus isolation was performed with the respective nasal swab samples, no conclusion could be drawn regarding the infectivity of the animals. The transmission of the infectious challenge virus to naïve animals of the same herd, thus, is a possible scenario and needs to be addressed in future studies where naïve contact animals are housed together with vaccinated and challenged animals.

Interestingly, seroconversion differed between both vaccination groups. All of the sheep vaccinated with the inactivated LSDV vaccine were positive for antibodies in ELISA and SNT on the day of challenge infection (Figure 4A,D). In contrast, on the day of challenge infection, the ELISA results were negative for all sheep vaccinated with the live-attenuated vaccine, and neutralizing antibodies could be detected in only two out of eight animals using SNT. Moreover, five animals remained negative in the ELISA during the entire study, and no neutralizing antibodies could be observed in any of the other sheep of this group until the end of the study (Figure 4B,E). Discrepancies between the sensitivity of the ELISA and SNT that were used are not surprisingly, as issues with the DA ELISA with sensitivity for samples derived from small ruminants have been previously observed [23,24]. Next to differences in the onset of seroconversion, differences in the neutralizing titer could be observed between inactivated LSDV-vaccinated animals and sheep that received the live-attenuated vaccine. Briefly, higher neutralizing titers could be observed after vaccination with the inactivated LSDV vaccine candidate, and titers increased during the challenge phase (Figure 4D). Strong seroconversion in this group is not unexpected as the tested inactivated LSDV vaccine candidate was able to induce high neutralizing antibody titers in cattle after vaccination in a previous study [40]. In the Lumpyvax-vaccinated sheep, the neutralizing titers were lower and did not increase during the days post-challenge infection (Figure 4B). The ELISA and SNT reactivities for the Lumpyvax-group B were basically very low. Thus, the results were consistent with each other. It can be assumed that the detected local replication did not lead to a robust systemic antibody response. Only for sheep S-300 was a robust serum neutralization was determined. Interestingly, these are the very sheep for which very low viremia was also detected. The observation that all vaccinated sheep were protected completely against the clinical signs of SPP after challenge infection with no correlation between protection and antibody titer supports previous reports that the immune status of LSDV-vaccinated or LSDV-infected animals cannot be related directly to the levels of neutralizing antibodies in the sera [3,55] and that cellular immunity also plays an important role in protection against CaPVs [56,57,58].

Not all scientific questions could be analyzed within the framework of this study. This concerns, in particular, the infectivity of the viruses detected in the nasal region of the vaccinated sheep as well as the possibility of virus transmission to other vaccinated and non-vaccinated animals. In addition, the cellular immune response after vaccination and challenge should be investigated more intensively in future trials.

## 5. Conclusions

In conclusion, both heterologous LSDV vaccines were able to completely protect sheep against strong a challenge infection with a highly virulent SPPV-India/2013/Surankote field strain. Although the local replication of the challenge virus could be observed in the vaccinated groups, both vaccines prevented a generalized infection and clinical signs with no exception. Because of its properties as an inactivated vaccine, our inactivated LSDV vaccine candidate, especially, could be an efficient and helpful tool for the prevention of SPPV infections as well as for the control of SPP outbreaks in the field in the future.

## Figures and Tables

**Figure 1 vaccines-11-00763-f001:**
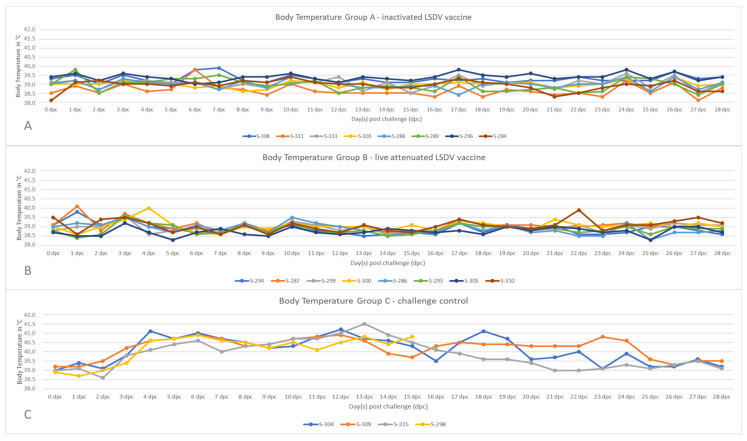
Body temperature after challenge infection. The body temperature of all animals was measured daily from 0 dpc to 28 dpc. Increased body temperature was defined ≥40.0 °C. (**A**) The animals of group A were vaccinated twice with an inactivated LSDV vaccine candidate. (**B**) The sheep of group B were vaccinated once with the live-attenuated LSDV vaccine strain. (**C**) The animals of group C served as the unvaccinated challenge control group.

**Figure 2 vaccines-11-00763-f002:**
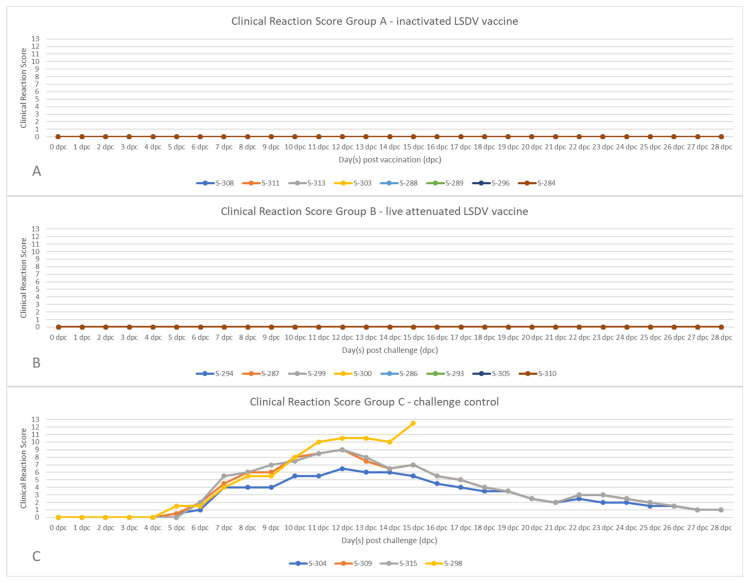
Clinical reaction score after challenge infection with the virulent SPPV-India/2013/Surankote. After the administration of the challenge virus, the development of clinical signs typical for SPP were examined and clinical reaction score was calculated. (**A**) The animals of group A were vaccinated twice with an inactivated LSDV vaccine candidate before challenge. (**B**) The sheep of group B received a single-shot of the live-attenuated LSDV vaccine strain before challenge. (**C**) The animals of group C served as the unvaccinated challenge control group.

**Figure 3 vaccines-11-00763-f003:**
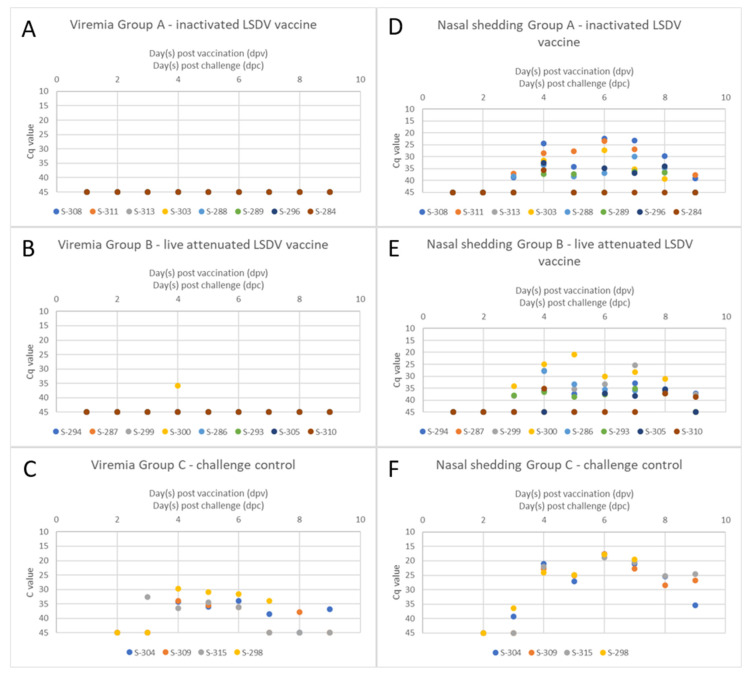
Viremia and viral shedding after challenge infection. (**A**–**C**) For the evaluation of viremia after challenge infection, EDTA blood samples were analyzed regarding their viral genome load. (**D**–**F**) Viral shedding was examined by viral genome load in the nasal fluid. (**A**,**D**) The animals of group A were vaccinated twice with an inactivated LSDV vaccine candidate before challenge. (**B**,**E**) The sheep of group B received a single-shot of the live-attenuated LSDV vaccine strain before challenge. (**C**,**F**) The animals of group C served as the unvaccinated challenge control group.

**Figure 4 vaccines-11-00763-f004:**
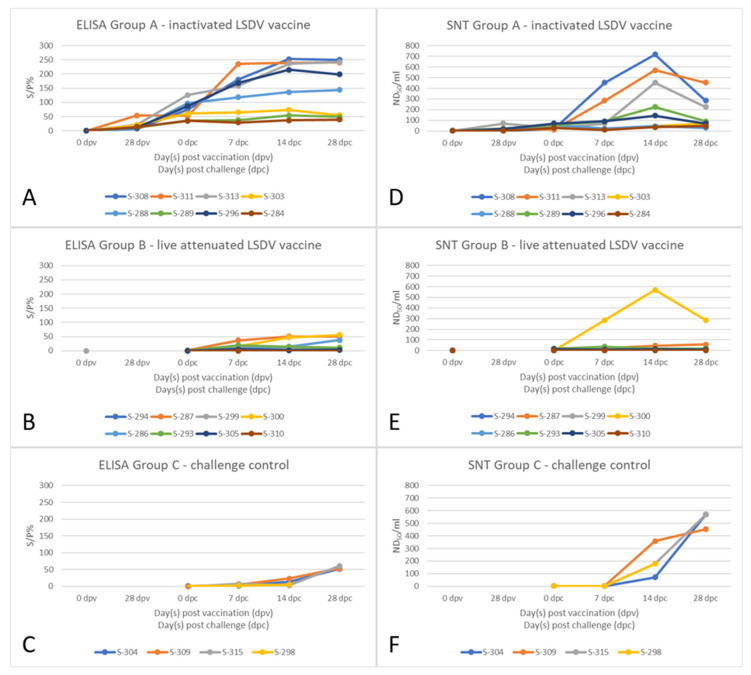
Serological response towards vaccination and challenge infection. (**A**–**C**) Overall serological response was measured using DA ELISA. (**D**–**F**) The neutralizing antibody titer was determined using the SNT. (**A**,**D**) The animals of group A were vaccinated twice with an inactivated LSDV vaccine candidate. (**B**,**E**) The sheep of group B were vaccinated once with the live-attenuated LSDV vaccine strain. (**C**,**F**) The animals of group C served as the unvaccinated challenge control group.

**Table 1 vaccines-11-00763-t001:** Viral genome load in the organ samples taken during necropsy. Cq value of the real-time PCR are presented; no Cq = no virus genome could be detected.

Sheep		Cervical Lymph Node	Mediastinal Lymph Node	Lung
Group A: inactivated LSDV vaccine	S-308	no Cq	no Cq	no Cq
	S-311	no Cq	no Cq	no Cq
	S-313	no Cq	no Cq	no Cq
	S-303	no Cq	no Cq	no Cq
	S-288	no Cq	no Cq	no Cq
	S-289	no Cq	no Cq	no Cq
	S-296	no Cq	no Cq	no Cq
	S-284	no Cq	no Cq	no Cq
Group B: live-attenuated LSDV vaccine	S-294	no Cq	no Cq	no Cq
	S-287	no Cq	no Cq	no Cq
	S-299	no Cq	no Cq	no Cq
	S-300	no Cq	no Cq	no Cq
	S-286	no Cq	no Cq	no Cq
	S-293	no Cq	no Cq	no Cq
	S-305	no Cq	no Cq	no Cq
	S-310	no Cq	no Cq	no Cq
Group C: challenge control	S-304	no Cq	no Cq	no Cq
	S-309	34.6	no Cq	no Cq
	S-315	no Cq	no Cq	no Cq
	S-298	34.4	no Cq	31.4

## Data Availability

The data presented in this study are available from the corresponding author upon request. The data are not publicly available due to funding by a third party.

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
