# Peer review of "Cross-Protection of an Inactivated and a Live-Attenuated Lumpy Skin Disease Virus Vaccine against Sheeppox Virus Infections in Sheep"

_vaccines, 2023, doi:10.3390/vaccines11040763_

Round 1

Reviewer 1 Report

The authors conduct a nice study comparing both a live attenuated LSDV vaccine and an inactivated LSDV vaccine to protect sheep against sheep pox. This is some new information that is publishable and the study is well written with the experimental results clearly demonstrating protection.

Minor corrections.

Line 337 it should be mentioned that these recombination events are caused by a vaccine containing multiple viruses. add reference Vandenbussche F, Mathijs E, Philips W, Saduakassova M, De Leeuw I, Sultanov A, Haegeman A, De Clercq K. Recombinant LSDV Strains in Asia: Vaccine Spillover or Natural Emergence? Viruses. 2022 Jun 29;14(7):1429. doi: 10.3390/v14071429. PMID: 35891412; PMCID: PMC9318037.

Line 339 This statement needs some additional clarification. Even with the use of capripoxvirus inactivated vaccines you can not demonstrate that vaccinated animals have not been exposed to capripoxvirus disease. So if a country required demonstrated proof by serology that the animals have not been exposed this can cause trade issues.

Line 338 Discuss the protection of sheep by a LSDV IL-10 knock out virus. add reference Boshra H, Truong T, Nfon C, Bowden TR, Gerdts V, Tikoo S, Babiuk LA, Kara P, Mather A, Wallace DB, Babiuk S. A lumpy skin disease virus deficient of an IL-10 gene homologue provides protective immunity against virulent capripoxvirus challenge in sheep and goats. Antiviral Res. 2015 Nov;123:39-49. doi: 10.1016/j.antiviral.2015.08.016. Epub 2015 Sep 1. PMID: 26341190.

Also discuss the use of the Kenyan sheep and goat pox vaccines which is a LSDV which have been used in the field as a vaccine to protect sheep and goats.

Line 417 One reason for the difference in antibody levels between the inactivated vaccine and the live attenuated vaccine is the number of vaccinations used. If you compare only a single vaccination the antibody levels will be similar between the inactivated and live attenuated vaccine.

Reviewer 2 Report

The manuscript is clear and interesting. I recommend the publication in the present form

Reviewer 3 Report

Overall comments

Sheeppox is an important agricultural pathogen. Safe and effective vaccines against sheeppox virus (SPPV) are needed. Inactivated vaccines have certain advantages, such as a lack of a potential for recombination or mutation, and avoiding introduction a live sheeppox infection into herds/regions/countries that do not have active infections in the animals. This paper shows that an inactivated whole virus vaccine gives more or less the same degree of clinical protection as a live attenuated vaccine, a useful finding.

Major

Overall, the writing needs to be improved. It is imprecise, and there are many usage and grammatical errors and typos.

Formal statistical tests need to be applied, even if some of the results look obvious by inspection.

It is disappointing that the ELISAs and neutralization tests were not shown against the strain used to challenge the animals.

Minor

The abstract is confusing and needs to be rewritten. The abstract needs to clearly make the point that there is a commercially available live attenuated vaccine against the cattle disease, lumpy skin disease, caused by the poxvirus lumpy skin disease virus (LSDV), which provides heterologous protection against SPPV, and that the authors have developed a new, inactivated LSDV vaccine that they are testing for its protective effects against SPPV. The abbreviation LSDV should be spelled out upon first use. The abstract needs to say why a new vaccine is needed, and how the current commercially available vaccine is lacking, and briefly why an inactivated vaccine would be better.. The abstract also needs to say that local nasal virus replication was detected after vaccination with both vaccines.

Introduction

Transmission via insects and arachnids is typically described as “vector” mediated, not “mechanical”. Line 34

What are “vaccine breakdowns”? line 62

The manuscript needs to be more precise in describing what is meant by “inactivated vaccine.” Does this mean killed whole virus vaccines or something different? How were the viruses in the vaccines inactivated? Were all the inactivated vaccines described in the introduction made in the same way?

What does “public opinion” mean in the sentence “Although public opinion claims less immunogenicity and induction of only short-lived protective immunity in animals vaccinated with inactivated CaPV vaccines [7],…” Reference 7 is 40 years old! Line 70

What is “strong challenge infection”? line 83

“Sterilizing immunity” not “sterile immunity” is the more commonly used term. Line 91

Methods

“Twenty-three to six-month old sheep” actually means sheep between 23 and 6 months old. Line 94

Why wasn’t the commercially available Lumpyvax vaccine used, instead of growing up home-made batches of LSDV?

What kind of swab was used to sample for viral shedding. The methods for the viral shedding assays really need to be described in more detail. How were the Cq values determined and calculated. What were the cycling conditions, and the instrument used? Were all the reagents, conditions, analysis software -- everything identical to those described in ref 6?

Results

Was there a statistically significant difference in temperature between the vaccinated groups and the control group? Between the two vaccinated groups?

Similarly, formal statistical tests should be applied to analysis comparing the different groups for all the parameters that were evaluated.

In Table 1, “no Cq” is really not the correct way of phrasing this negative result.

The results presented in Fig 4 really don’t make much sense. Was there really no ELISA titers developed by the sheep in the control group? Shouldn’t they have made a potent immune response, since they recovered from the infection. In another difficult to explain result, why did the live attenuated vaccine group not develop neutralizing antibodies. It looks like the ELISAs and the neutralizing antibody assays were run using the vaccine strain. However, the really important ELISA and neutralizing results would be against the challenge strain. This data should be presented.

Conclusions/Discussion

Since the necropsies were done at the conclusion of the study, it really isn’t possible to say that the internal organs were not infected at all, just that virus was effectively cleared in the vaccinated animals.

The Discussion should include a more thorough discussion of the limitations of this study.

How does the dose of the live virus vaccine as administered in this study compare to the dose of the licensed live attenuated vaccine?

Figures and Tables

It would be better in Fig 3 A, B, C if the y-axis was not so compressed. These are really different assays than D, E, F, and so there is no need to have the scales be the same.

Round 2

Reviewer 3 Report

The manuscript is acceptable.